# DNA methylation in blood—Potential to provide new insights into cell biology

**Donia Macartney-Coxson**[1]*, **Alanna M. Cameron**[2], **Jane Clapham**[1], **Miles C. Benton**[1]*

**1** Human Genomics, Institute of Environmental Science and Research (ESR), Porirua, Wellington, New Zealand, **2** Malaghan Institute of Medical Research, Wellington, New Zealand

* miles.benton84@gmail.com, miles.benton@esr.cri.nz (MCB); Donia.Macartney-Coxson@esr.cri.nz (DMC)

**Data Availability Statement:** All data (raw and processed) and results are available via GitHub (https://github.com/sirselim/immunecell_

## Abstract

Epigenetics plays a fundamental role in cellular development and differentiation; epigenetic mechanisms, such as DNA methylation, are involved in gene regulation and the exquisite nuance of expression changes seen in the journey from pluripotency to final differentiation. Thus, DNA methylation as a marker of cell identify has the potential to reveal new insights into cell biology. We mined publicly available DNA methylation data with a machine-learning approach to identify differentially methylated loci between different white blood cell types. We then interrogated the DNA methylation and mRNA expression of candidate loci in CD4+, CD8+, CD14+, CD19+ and CD56+ fractions from 12 additional, independent healthy individuals (6 male, 6 female). 'Classic' immune cell markers such as CD8 and CD19 showed expected methylation/expression associations fitting with established dogma that hypermethylation is associated with the repression of gene expression. We also observed large differential methylation at loci which are not established immune cell markers; some of these loci showed inverse correlations between methylation and mRNA expression (such as *PARK2*, *DCP2*). Furthermore, we validated these observations further in publicly available DNA methylation and RNA sequencing datasets. Our results highlight the value of mining publicly available data, the utility of DNA methylation as a discriminatory marker and the potential value of DNA methylation to provide additional insights into cell biology and developmental processes.

## Introduction

Epigenetics refers to the heritable, but reversible, regulation of various genomic functions, including gene expression. It provides mechanisms whereby an organism can dynamically respond to changes in its environment and "reset" gene expression accordingly [1]. Furthermore, these mechanisms play a critical role in development and cell lineage specificity [2, 3], as highlighted recently when epigenomic profiling revealed a linear differentiation model for memory T-cells [4]. One such epigenetic mechanism is DNA methylation. Methylation of the cytosine nucleotide within CpG dinucleotides in DNA is well documented in humans [5, 6]. DNA methylation can be developmentally 'hard-wired' (as in the case of imprinting [7]), underpin cell identity (i.e. cell markers of differentiation [8, 6]) or dynamic and change in response to environmental factors [9]. Therefore, the investigation of an individual's

methylation_paper_data)[DOI:10.5281/zenodo.336694].

**Funding:** The study was supported in part by funding provided by the Wellington Medical Research Foundation (https://researchforlife.org.nz/) grant number 2015/255 awarded to MB. The funders had no role in the study design, data collection and analysis, decision to publish or preparation of the manuscript. There was no additional external funding received for this study.

**Competing interests:** The authors have declared that no competing interests exist.

methylation pattern can reveal a lifetime record of environmental exposures as well as potential disease specific marks [10, 11].

It is well established that epigenetics contributes significantly to the developmental fate of cells and tissues [8]. For instance, epigenetic mechanisms contribute to the differentiation of hematopoietic stem cells from bone marrow [12, 13]. Importantly, DNA methylation appears to play a crucial role at specific stages along the separation of blood cell lineages (myeloid, lymphoid) and contributes to the establishment and functionality of the final differentiated cell type [14]. Epigenetic marks, including DNA methylation, are increasingly recognised as potential discriminators of cell type [15]. This attribute has been utilised by a number of researchers to develop methods which correct for and/or deconvolute the variability introduced by cell mixtures in DNA methylation studies, particularly in blood samples [16–20]; a notable example—the so-called Houseman algorithm (Houseman 2012)—has been incorporated in to standard bioinformatic pipelines, including the R minfi package [21], for DNA methylation arrays. This behaviour of DNA methylation as a marker also suggests the possibility of such 'marks' revealing new aspects of biology—for instance it may highlight previously unrecognised immune cell populations.

DNA methylation as an epigenetic mark is easily quantified and evaluated from blood. Many studies using Illumina array technology have made their data publicly available, providing an excellent resource for hypothesis generation and testing *in silica* prior to wet-lab experimentation. We hypothesised that because of its role in differentiation and development new biological insights could be revealed by looking at loci that discriminate between immune cell types; the potential utility of these loci in cell discrimination might be previously unrecognised and/or could be harnessed to sort and/or identify potential new cell sub-types. Therefore, we performed an *in silico* discovery experiment using data from a study which examined the DNA methylation profile of human white blood cell populations [22]. Reinius *et al.*, investigated DNA methylation in: T cells (CD8$^+$, CD4$^+$); B cells (CD19$^+$); natural killer cells (NK cells; CD56$^+$); monocytes (CD14$^+$); granulocytes (Gran; both CD16$^+$ and Siglec8$^+$ cells); neutrophils (Neu, CD16$^+$), and eosinophils (Eos, Siglec-8$^+$). The Reinus study was one of the first to illustrate the potential power of DNA methylation as a biomarker, and its role in cell lineage identity with the authors profiling DNA methylation in six healthy males and identifying discriminatory DNA loci in "classic" immune cell marker loci. Here, we use a machine learning approach which, as anticipated, identifies discriminatory DNA methylation marks in 'classic' immune cell markers, but also highlights significant differential methylation in "non-classic" immune markers, and genes for which a role in immune function is yet to be reported. We interrogate this further in an independent cohort and publicly available data at both the DNA methylation and gene expression level.

## Material and methods

### Discovery analysis

**DNA methylation analysis.** The Reinus data was downloaded using the R package MAR-MAL-AID [23]. All applicable sample information is available at the GEO page (GSE35069, https://www.ncbi.nlm.nih.gov/geo/query/acc.cgi?acc=GSE35069).

Raw intensity data (Illumina 450K idats) were loaded into R [24] using the Bioconductor minfi package [21]. Background correction and control normalisation was implemented in minfi. Probes were classed as failed if the intensity for both the methylated and unmethylated probes was <1,000. Any probe which failed in at least one sample, was removed from the entire dataset. We also removed all previously identified cross-reactive probes [25], and 33 457 probes which we previously identified as aligning to the human genome greater than once

[26]. All analyses were performed on beta values, calculated as the intensity of the methylated channel divided by total intensity including an offset ((methylated + unmethylated) + 100).

Glmnet penalised ridge-regression mixed with lasso in an elastic-net framework was used as implemented via the R package glmnet [27] to explore methylation association between each of the cell-types (CD8+, CD4+, CD19+, CD14+, CD56+, Neutrophils, Eosinophils, Granulocytes, as well as combinations of cell populations, PBMC and whole blood). The number of variables (~450,000 CpG sites, Illumina 450K platform) far outweighs the number of cell-types; as such it is accepted that conventional statistical analysis procedures that test each CpG within an independent regression model suffer from multiple testing burden and reduced statistical power. To overcome this issue we chose to use the penalised regression procedures of glmnet, which tests all markers simultaneously, i.e. in a single regression model. Glmnet was specifically designed to overcome issues of large variable number (k) and small sample size (n) and has been successfully applied to several genome-wide association studies of SNPs [28–30] and recently methylation [31]. We have previously developed and reported on this method in detail to identify aging associated DNA methylation loci [26]. The Flt-SNE software with associated R wrapper function was used for t-SNE analysis [32]. Briefly, glmnet fits a generalized linear model via penalized maximum likelihood. The regularization path is computed for the lasso or elastic-net penalty at a grid of values for the regularization parameter lambda $\lambda$. The elastic-net penalty is controlled by $\alpha$, and bridges the gap between lasso ($\alpha = 1$, the default) and ridge ($\alpha = 0$). The ridge penalty shrinks the coefficients of correlated predictors towards each other while the lasso tends to pick one of them and discard the others. The elastic-net penalty mixes these two; if predictors are correlated in groups, an $\alpha = 0$ tends to select the groups in or out together. We selected an alpha at the lower end of the range (0.05) to shift the elastic-net model more towards the penalised-regression (ridge regression), allowing us to retain more related features (CpG sites which share variance). For the glmnet modelling we used cross-validation to determine the optimal value of regularization parameter $\lambda$ with both minimum mean squared error (MSE) and minimum MSE + 1SE of minimum MSE. The optimal $\lambda$ values were then used for predictor variable selection.

**Pathways enrichment.** Functional enrichment was performed on each set of CpG sites identified for each cell type in the ToppGene Suite webserver (https://toppgene.cchmc.org/) using the ToppFun function. Bonferroni adjusted correction was used in the reporting of all pathways results (adjusted $P < 0.05$).

## Validation analyses

**Samples.** Ethics was obtained from, and all experimental protocols were approved by, The Health and Disability Ethics Committee NZ (HDEC, 15/NTB/153). All methods were carried out in accordance with relevant guidelines and regulations. Written, informed consent was obtained from all participants who were all over 18 years of age at the time of collection. Blood from 12 self-reported healthy individuals (n = 6 male, n = 6 female) between 26–31 years of age inclusive, was collected into sterile K2 EDTA vacutainers (BD Biosciences), and the buffy coat isolated.

**Cell sorting–FACS.** Peripheral blood mononuclear cells (PBMCs) were Fc receptor blocked, labelled with fluorescent antibodies specific for: CD3 (OKT3), CD4 (OKT4), CD8 (HIT8a), CD14 (HCD14), CD19 (HIB19) and CD56 (HCD56; all antibodies were from Biolegend) and dead cells were identified by DAPI exclusion. CD4+, CD8+, CD14+, CD19+ and CD56+ fractions were collected (Influx cell sorter, BD Biosciences) directly into ice-cold FACS buffer, immediately frozen on dry ice and stored at –80˚C.

**DNA and RNA extraction.** Both nucleic acids were extracted simultaneously from snap frozen cells using a Qiagen All prep DNA/RNA kit as per the manufacturers protocol. High

**Table 1. Annotation, methylation status and TaqMan probe information for the 11 selected CpG sites.**

| IlmnID | CellType | Cell meth (mean) | Other meth (mean) | Absolute Difference | Percent Difference | CHR | Position | Gene Symbol | Feature | TaqMan Probe |
|---|---|---|---|---|---|---|---|---|---|---|
| cg24462702 | CD4 | 0.13 | 0.82 | 0.69 | 69.07 | X | 135730445 | *CD40LG* | 1stExon | Hs00163934_m1 |
| cg10837404 | CD4 | 0.34 | 0.88 | 0.54 | 54.14 | 5 | 112356289 | *DCP2* | 3'UTR | Hs00400339_m1 |
| cg02665297 | CD19 | 0.08 | 0.95 | 0.87 | 86.91 | 7 | 5270984 | *WIPI2* | 3'UTR | Hs01093807_m1 |
| cg21596498 | CD19 | 0.12 | 0.92 | 0.8 | 80.33 | 19 | 42618407 | *POU2F2* | Body | Hs00922179_m1 |
| cg27565966 | CD19 | 0.12 | 0.87 | 0.75 | 74.94 | 16 | 28943198 | *CD19* | TSS200 | Hs01047412_g1 |
| cg25939861 | CD8 | 0.14 | 0.81 | 0.67 | 67.35 | 2 | 87020937 | *CD8A* | 5'UTR | Hs01555594_g1 |
| cg11067179 | CD8 | 0.41 | 0.84 | 0.43 | 42.85 | 11 | 66083541 | *CD248* | 1stExon | Hs00535586_s1 |
| cg23244761 | CD14 | 0.14 | 0.93 | 0.79 | 78.76 | 6 | 161796850 | *PARK2* | Body | Hs01038322_m1 |
| cg16636767 | CD14 | 0.21 | 0.89 | 0.69 | 68.55 | 11 | 13694647 | *FAR1* | 5'UTR | Hs00386153_m1 |
| cg13617280 | CD56 | 0.25 | 0.88 | 0.63 | 63.42 | 12 | 129299462 | *SLC15A4; MGC16384* | Body; TSS200 | Hs00377326_m1 |
| cg13995453 | CD56 | 0.43 | 0.88 | 0.45 | 45.19 | 12 | 9759653 | *KLRB1* | Body | Hs00174469_m1 |

quality genomic DNA and RNA were obtained, with RNA RIN $\geq$ 7.5. Sufficient quality and quantity of DNA and RNA was obtained to facilitate targeted DNA methylation and mRNA expression profiling for CD4+, CD8+, CD19+, CD14+ and CD56+ cell sorted samples.

**Targeted DNA methylation analysis.** Pyrosequencing was designed and performed by EpigenDX (USA), who were provided with the Illumina probe information (Table 1).

**Targeted gene expression analysis.** 150ng total RNA was reverse transcribed using VILO Superscript (Thermo Fischer). QRTPCR was performed in triplicate on 7ng cDNA using TaqManGene expression assays (*CD40LG* Hs00163934_m1, *DCP2* Hs00400339_m1, *WIPI2* Hs01093807,*POUF2* Hs00922179_m1, *CD19* Hs01047412_g1, *CD8A* Hs01555594_g1, *CD248* Hs00535586_s1, *PARK2* Hs01038322_m1, *FAR1* Hs00386153_m1, *SLC15A4* Hs01547421_m1, *KLRB1* Hs00174469_m1). Gene expression was normalised against the non-variable endogenous control genes *GAPDH* and *GUSB*, using the $\Delta$Ct method ($Ct_{candidate}$-$MeanCt_{controls}$).

## Statistics

All analyses were performed in R 3.5.2. Differential methylation and expression analyses were performed in R using the default student t-test. P values were adjusted using the Benjamini-Hochberg method.

## Data

All raw and processed data are accessible via GitHub, see https://github.com/sirselim/immunecell_methylation_paper_data [DOI:https://doi.org/10.5281/zenodo.3366393]. A github repository and related site have been made available to explore t-SNE results interactively (https://sirselim.github.io/tSNE_plotting/).

## Results

### Discovery—DNA methylation discriminatory markers for immune cells

We identified DNA methylation at 1173 CpG sites (S1 Table) which clearly differentiated specific immune cell populations using publicly available data from whole blood [16]; hierarchical clustering and t-SNE analyses provide a visual presentation and highlight that these markers

cluster the cell populations in a biologically meaningful way (Fig 1). Pathway analyses of the genes to which these 1173 CpG sites mapped strongly supported their discriminatory nature, and, as expected, enrichment for immune cell biological function was observed: enrichment for CD56 (> 79 genes), CD4 (> 68 genes), CD8 (> 34 genes), CD14 (> 69 genes) and CD19 (> 194 genes) was observed. Furthermore, these results suggest that discriminatory CpG marker loci may map to genes with a hitherto unrecognised role in immune cell discrimination and/or function.

The robust differentiation between cell types was explained by non-overlapping sets of CpGs specific for each cell population: CD8+ (n = 70); CD4+ (n = 96); CD19+ (n = 347); CD56+ (n = 112); CD14+ (n = 126); Granulocytes (n = 128); Neutrophils (n = 128), and Eosinophils (n = 166). The majority of these sites were relatively hypo-methylated in the cell type of discrimination and hyper-methylated in all other cell populations analysed. The proportion of hypomethylated/total non-overlapping discriminatory CpGs [for a given cell type] was: CD8+ (46/70, 65.7%), CD4+ (71/96, 74%), CD19+ (344/347, 99%), CD56+ (111/112, 99%), CD14+ (126/126, 100%), Granulocytes (94/128, 73.4%), Eosinophils (165/166, 99%) with Neutrophils being the exception (33/128, 24.2%).

Interestingly, the majority of CpG marker sites identified (~95% of CpGs) mapped to annotated gene loci, with many in regions involved in regulating mRNA expression (e.g.

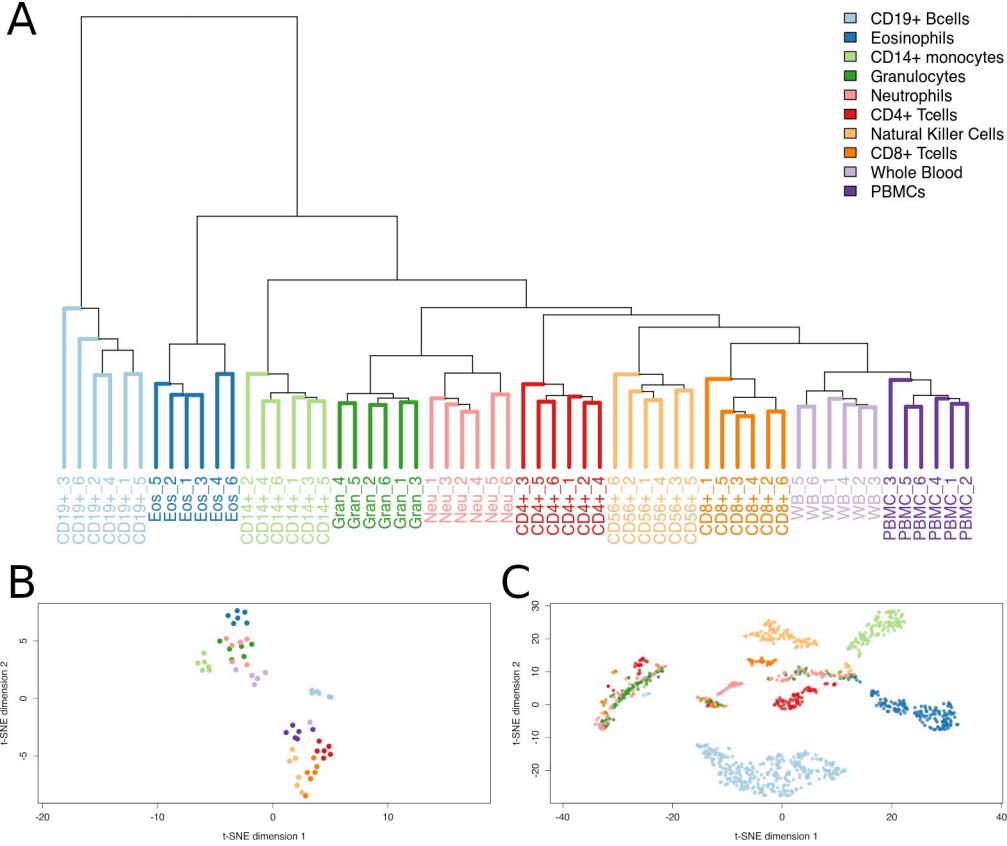

**Fig 1. Demonstration of immune cell population discrimination using sets of identified epigenetic markers (CpGs).** A) Hierarchical clustering of all 1173 identified probes demonstrates perfect separation of cellular populations. B) Plot of t-sne dimensions derived from all methylation sites for all 60 samples. Points on the plot represent individual samples. C) 2D t-sne plot of selected 1173 methylation markers identified via glmnet method. Points on the plot represent individual CpG sites. An interactive version of this panel is available (https://sirselim.github.io/tSNE_plotting/).

promoters). For each cell type marker the proportion of CpG sites mapping to annotated loci was: CD8+ (62/70); CD4+ (78/96); CD19+ (255/347); CD56+ (99/112); CD14+ (82/126); Granulocytes (102/128); Eosinophils (136/166), and Neutrophils (108/128). For individual marker information including annotation see https://github.com/sirselim/immunecell_methylation_paper_data [DOI: https://doi.org/10.5281/zenodo.3366393].

The largest DNA methylation difference observed was 87% between CD19+ cells against all others. This 87% difference was observed in two genes, *WIPI2* and *CARS2;* while *WIPI2* has a reported role in the immune system [33], to the best of our knowledge no such function has been reported for *CARS2* to date. Ranked by the largest change in methylation the top five CpG sites mapping to annotated loci for each cell type were:

CD19+: 87% (*WIPI2, CARS2*), 83% (*RERE*), 82% (*LOC100129637*), 80% (*POU2F2*)

CD4+: 69% (*CD40LG*), 56% (*PUM1*), 54% (*DCP2, BAG3*), 48% (*SF1*)

CD8+: 67% (*CD8A*), 64% (*CD8A*), 51% (*CD8B*), 49% (*CD8B, CD8A*)

CD56+: 63% (*SLC15A4*), 52% (*RASA3*), 48% (*MAD1L1*), 45% (*KLRB1/CD161*), 43% (*KLRB1/CD161*)

CD14+: 79% (*PARK2*), 70% (*CENPA, PARK2*), 69% (*KIAA0146, FAR1*)

Eosinphils: 73% (*FAM65B*), 72% (*KIAA0317, APLP2*), 70% (*MEF2A, CCDC88A*)

Granulocytes: 60% (*VPS53, PCYOX1*), 59% (*ARG1*), 58% (*CSGALNACT1*), 56% (*SH3PXD28*)

Neutrophils: 14% (*CUL9*), 12% (*LASP1*), 7% (*GFl1*), 6% (*LRFN1, NFAT5*)

## Validation in independent samples

In order to validate our observations from the *in silica* discovery experiment we selected 11 differentially methylated loci (Table 1) for analysis in 12 independent samples from self-reported healthy individuals (n = 6 female, n = 6 male) with an age range of 26–31 years inclusive. This sample size is equivalent per sex to that of the Reinius data [22] used in the discovery analysis.

Our validation concentrated on cell sorted populations for CD4+, CD19+, CD4+, CD8+, CD56+, CD14+ from which it was possible to collect enough cells for simultaneous extraction of DNA and RNA of sufficient quantity and quality.

Ten loci were selected for validation, two for each cell type (Table 1). The most differentially methylated site for each cell type CD4+, CD19+, CD8+, CD56+, CD14+ was selected (*WIPI2, CD40LG, CD8A, SLC15A4, PARK2* respectively). A second site from the top 5 (see above) was selected for each of CD19+, CD4+, CD56+ and CD14+ (*POUF2, DCP2, KLRB1, FAR1*). For CD8+ all sites in the top 5 mapped to this marker, we therefore selected the sixth top loci which mapped toCD248 (43% difference in methylation). In addition, CD19 (ranked 27th in terms of % differential methylation [75%] of annotated loci) was included as a control.

**DNA methylation.** The eleven candidate loci were assayed by pyrosequencing in the 12 samples from the validation cohort. We observed a strong agreement with the expected discriminatory patterns of DNA methylation for all loci examined (Figs 2 and 3). S2 Table presents pair-wise student T-test statistics for the DNA methylation data.

**RNA expression.** Given the role that DNA methylation plays in regulation of gene expression we also explored the mRNA levels of the 11 candidate loci. We investigated gene expression by QRTPCR in the 12 independent, validation samples. A clear differentiation between immune cells types at the gene expression level was observed for PARK2, POU2F2, DCP2, CD248, CD8A, SLC15A4, CD4A0LG and CD19 but not for FAR1, WIPI2, KLRB1 (Figs 2 and 3). S3 Table presents pair-wise student T-test statistics for the gene expression data.

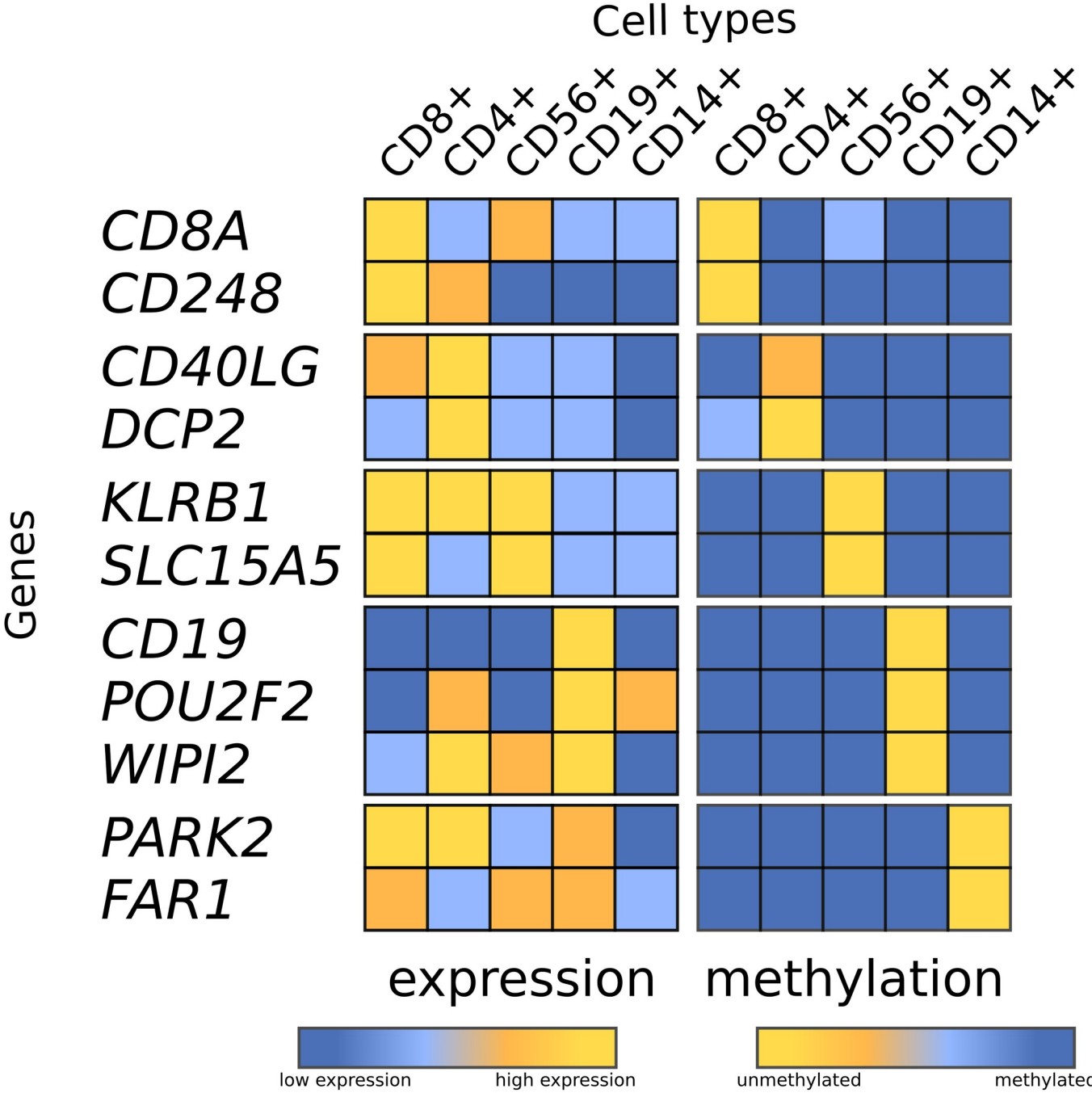

**Fig 2. Heatmap representation of DNA methylation and gene expression data for all 11 genes investigated.** Expression and methylation measures were split into quartiles and their levels coloured accordingly.

### Validation in publicly available data

**DNA methylation.** In order to further investigate the panel of 1173 CpG sites identified in our initial analysis we interrogated their methylation in 3 publicly available data sets. One, GSE82084, using the Illumina 450K platform (as per the Reinus data used in our discovery analysis) and two (GSE103541, GSE110554) using the more recent Illumina EPIC platform. Of

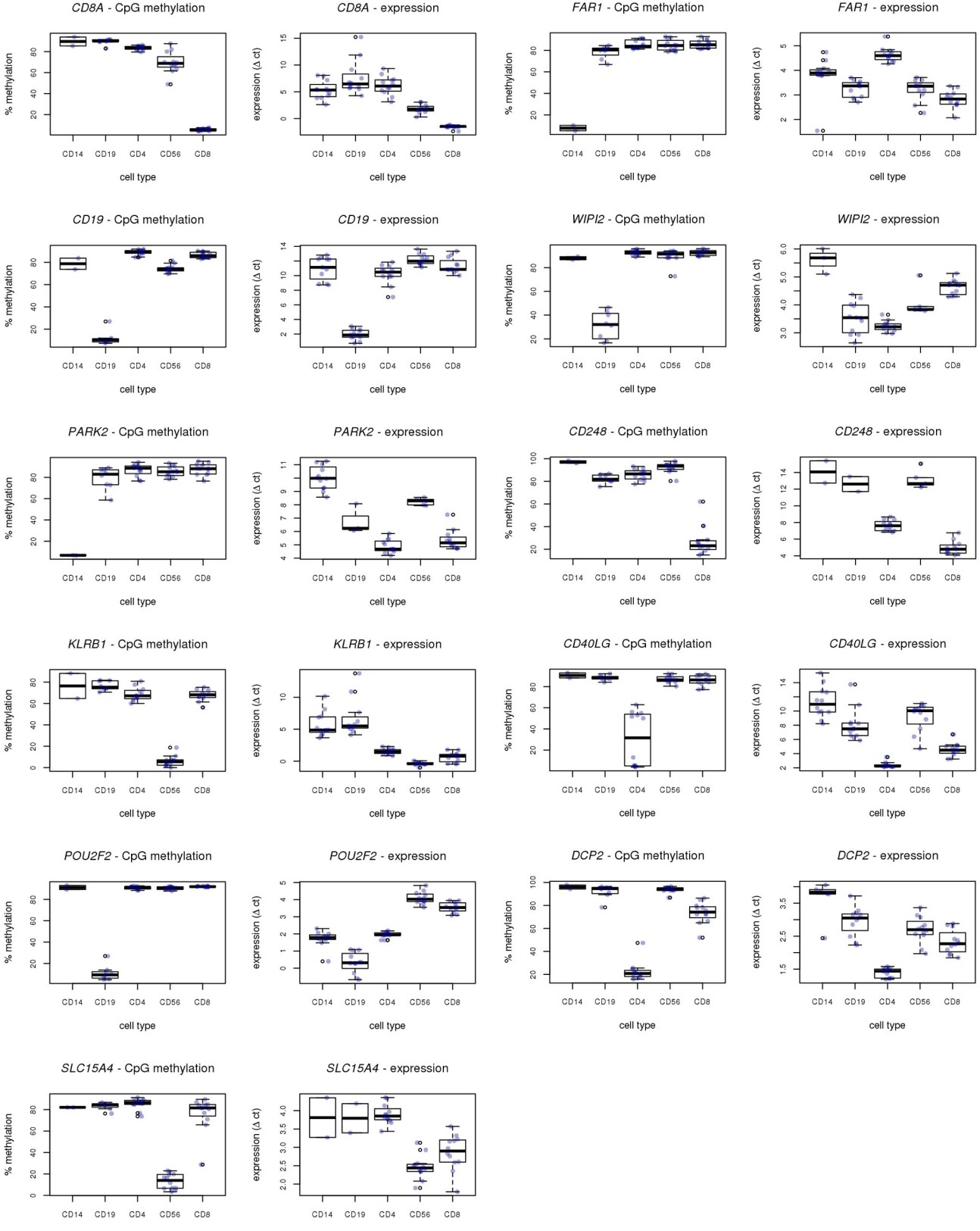

**Fig 3. Boxplots illustrating DNA methylation and gene expression levels for all 11 gene investigated.** Methylation and gene expression data for a given gene are in adjacent boxplots.

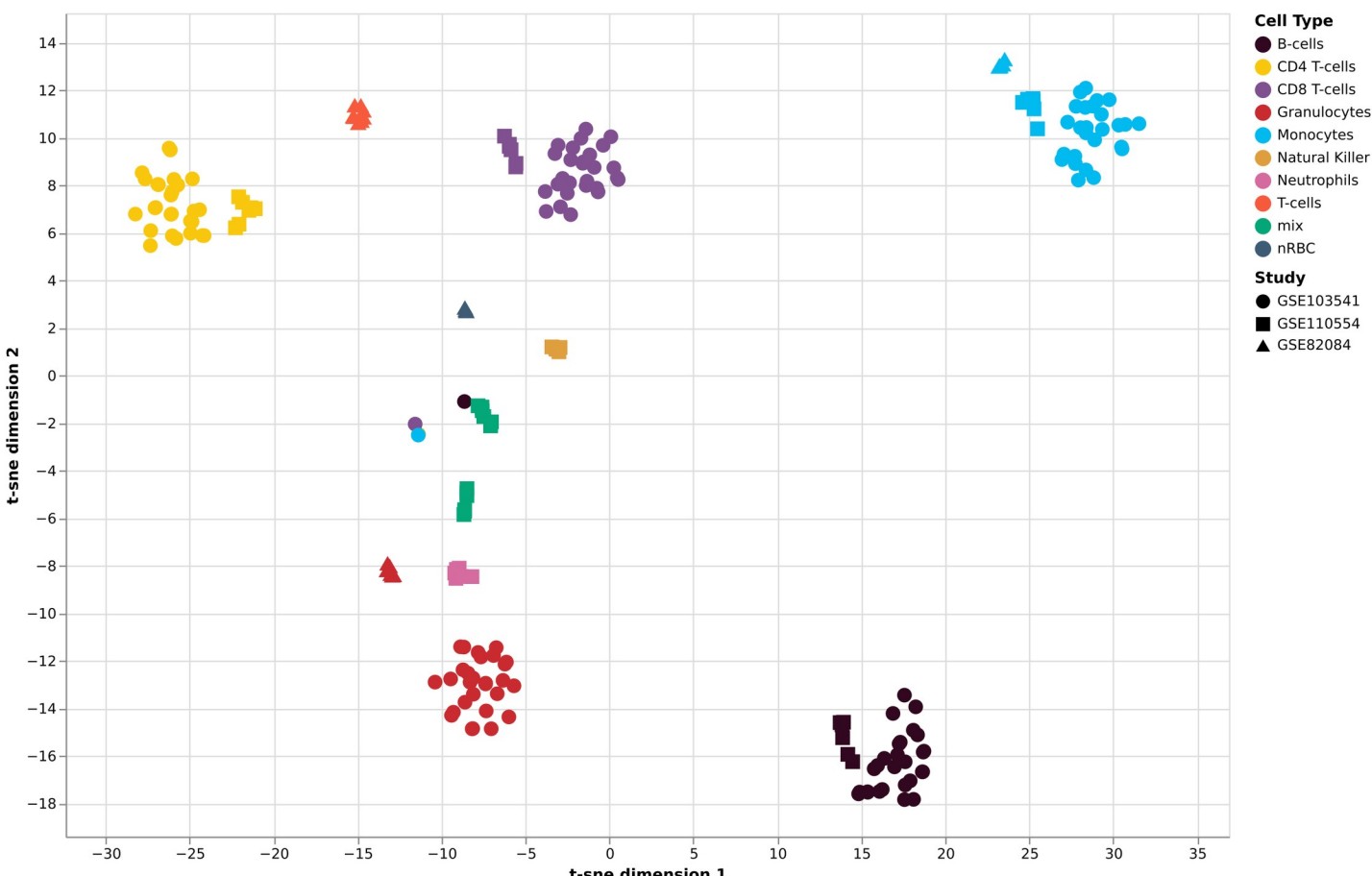

**Fig 4. tSNE plot of sorted-cells from 211 samples based on 1025 CpG sites overlapping between the three publicly available datasets.** Points represent individual samples.

the 1173 CpG sites 1025 were present on both platforms. The two EPIC studies performed DNA methylation analysis of cell sorted immune cell populations from adults [17], whereas the 450K study looked at DNA methylation in cord blood from term and preterm newborns [34]. Fig 4 presents a 2D tSNE plot of all 1025 CpG sites which clearly shows separation of immune cells populations. It is interesting to note that the T cells of neonates (orange/red triangles) sit between the CD4+ and CD8+ T cells consistent with an undifferentiated state. We also observed independent clustering of nucleated red blood cells from the same preterm newborns cohort, despite the fact that this cell-type was not in our training set.

**RNA expression.** To further explore expression of the 11 genes we selected for our validation of DNA methylation and RNA expression on independent samples we interrogated a publicly available RNAseq dataset, GSE107011 [35]. We extracted data for the same five cell populations (CD4+, CD8+, CD19+, CD56+ and CD14+) and then extracted expression data (TPM, transcripts per million) for each of the 11 genes. Fig 4 presents the data and illustrates the ability of mRNA expression from these 11 genes to clearly differentiate cell types. It is interesting to note that effector memory/terminal effector CD8+ cells (purple circles) are distinguished from central memory/naive CD8+ cells (purple squares) as well as a separation of Th17 CD4+ cells (yellow diamonds) from other CD4+ cells (yellow circles). Interactive versions of all figures are available online (https://sirselim.github.io/tSNE_plotting/).

## Discussion

DNA methylation is exquisitely placed to reflect a cell's differentiation trajectory. Using publicly available data, and a machine learning approach we identified 1173 unique CpG sites at which DNA methylation discriminated CD8+, CD4+, CD19+, CD56+, and CD14+ cell populations as well as granulocytes, neutrophils, and eosinophils. We validated DNA methylation in two discriminatory CpG loci for each of CD8+, CD4+, CD19+, CD56+, and CD14+ in 12 independent samples.

The majority of the 1173 discriminatory CpG sites mapped to annotated loci, and gene regulatory regions in particular. This suggests that, as expected, DNA methylation is playing a key role in immune cell differentiation and cell-type identification. An important implication of this is that DNA methylation can therefore be potentially harnessed to reveal previously unidentified aspects of biology, such as immune cell sub-populations. A good example of this is the transcription factor FOXP3 which plays a key role in the development and function of Treg cells [36]; originally FOXP3 expression was used to identify Treg cells until it was deemed insufficient for the robust identification of suppressive Treg cells [37, 38]. However, recent work has reported that hypomethylated CpG sites in four regions of FOXP3, CAMTA1 and FUT7 can be used to distinguish subsets of Tregs from non-regulatory CD4+ T cells [39]. These findings strongly support our view that DNA methylation, including potentially loci identified in the current study, could be used to inform similar experiments and reveal other drivers of specific immune cell subtypes.

Furthermore, large differences in DNA methylation were observed, and validated, at CpG loci in genes which, while their potential role in immune cell biology has been reported, have not previously been recognised as differentiators of immune cell type, such as *WIPI2* [33] for CD19+,*SLC15A*4 [40–42] for CD56+ and *PARK2* [43, 44] for CD14+ cells. We also identified *POUF2/OCT2* for which a role as a B-cell differentiator was recently reported [14]. In addition, significant, cell type specific changes in DNA methylation were observed, and validated, in genes which, to the best of our knowledge, have no previous reported role in immune biology (FAR1, CARS2). Taken together this highlights the significant potential of such analyses to uncover new facets of cell biology, and immunology. Many more additional loci from our *in silica* analyses showed large differences in DNA methylation, and these warrant further investigation with respect to their roles in immune cell function.

To further explore the potential relationships between our selected cell methylation markers we used a t-SNE; a statistical method that attempts to identify higher dimensional relationships between data points and assign a faithful representation of those points in lower dimensional space (usually 2D) [45]. As a method t-SNE has been widely adopted in single cell sequencing experiments to identify clusters of cell populations [46, 47]. The resultant t-SNE analysis of the selected 1173 markers (Fig 1) clearly demonstrates distinct groupings of CpG sites into respective cell populations. There is a small degree of non-specific clustering of CpG sites. This could well be due to higher order background 'signal', or it could potentially be pointing towards underlying biological relationships that have yet to be established. The potential of this approach is highlighted by our analyses of publicly available data and 1025/1173 'candidate CpG sites' which overlapped between 450K and EPIC Illumina bead platforms. Fig 4 illustrates how well these 1025 CpG sites performed in additional, independent data. Furthermore, our initial analyses focused on samples from adults and as such we could not comment on their performance in neonates. However, one of the three public datasets we explored was from a study investigating DNA methylation in cord blood from term and preterm newborns. This clearly shows separation of immune cell sub-types isolated from neonates with the CpG markers we identified. It also suggests that such 'biomarkers' can potentially identify additional

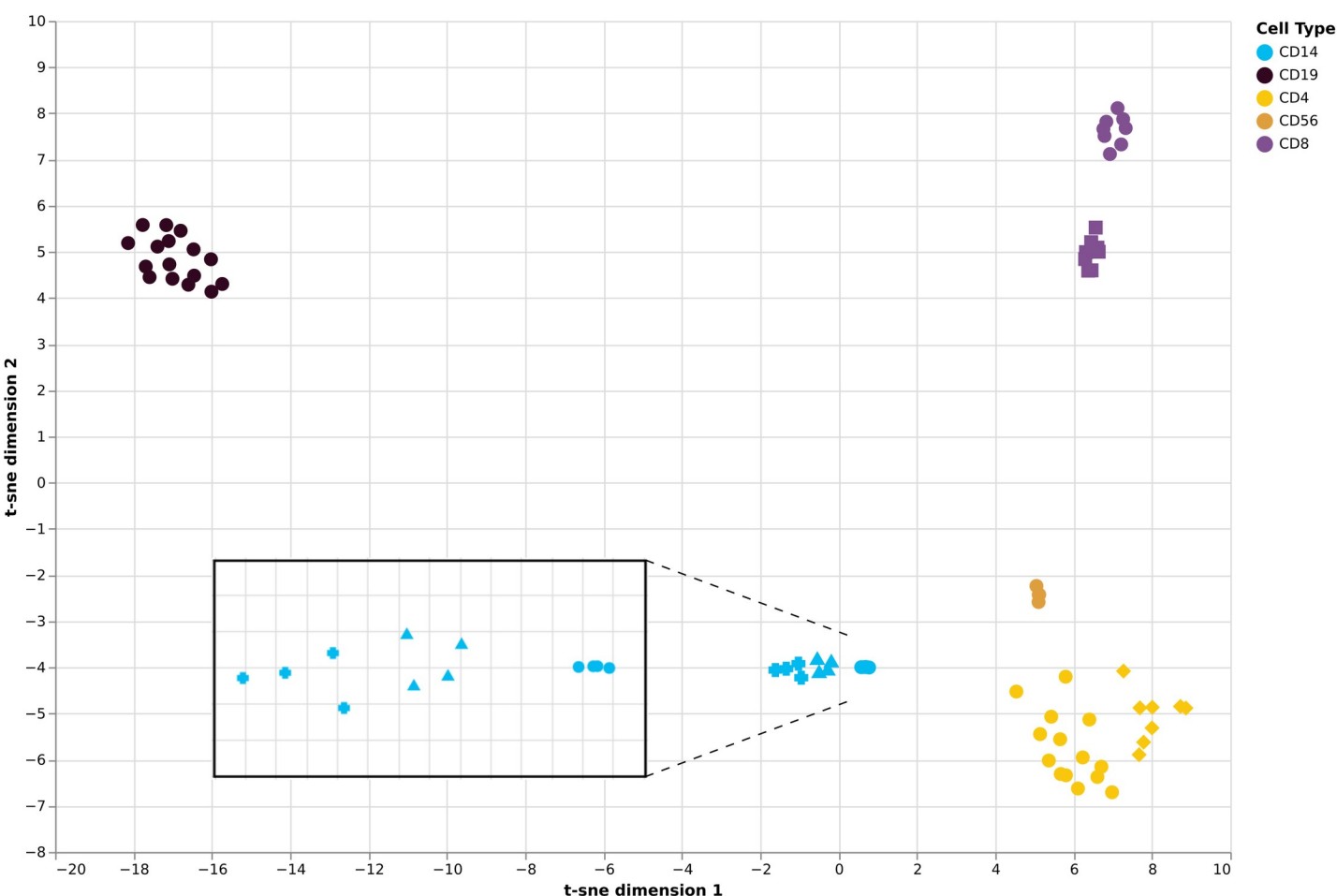

**Fig 5. tSNE plot of RNA expression in publicly available data for sorted-cells for the 11 genes highlighted in this study (RNAseq data from GSE107011).** Points represent individual samples. CD8+ central memory/naive purple squares, CD8+ effector/memory/terminal effector (purple circles), CD4 Th17 cells (yellow diamonds), other CD4 T-helper cells (yellow circles), CD14 monocytes—classical (blue circles), intermediate (blue triangles), non-classical (blue crosses). Interactive versions of all figures are available online (https://sirselim.github.io/tSNE_plotting/).

aspects of cell identity; for instance, the T cells of neonates (orange/red triangles) sit between the mature CD4+ and CD8+ T cells consistent with an undifferentiated state. Furthermore, we also observed independent clustering of nucleated red blood cells from the same preterm new-borns cohort. We believe these observations support the tantalising possibility that DNA methylation can be harnessed to reveal new aspects of cell biology including the identification of currently unrecognised/undistinguishable immune cell sub-types.

mRNA expression analysis of the genes to which the 11 validated DNA methylation discriminatory loci mapped also revealed discrimination at the mRNA level for *CD248*, and *CD8A* (CD8+), *POU2F2* and *CD19* (CD19+), *PARK2* (CD14+), *DCP2* (CD14+), *SLC15A4* (CD56+), and *CD40LG* (CD4+). There were three genes (*FAR1*, *WIPI2*, *KLRB1*) for which this was not observed. One potential explanation is the presence of multiple isoforms per gene, such that the primer/probe combination for the QRTPCR analysis did not target the correct isoform. This possibility warrants further investigation especially given the increasing body of evidence that DNA methylation is an important modulator of alternative splicing [48–50]. We also investigated the expression of the 11 genes in a publicly available RNAseq dataset from immune cell sorted populations and saw a clear separation of the cell types with these 11

transcripts (Fig 5). In addition, the t-SNE analysis hints at the power of these 11 transcripts to provide a more nuanced separation of cell types. For example, we observed distinct separation of CD8 T-cells into two clusters of sub-populations (Terminal Effector/Effector Memory and Central Memory/Naive). Similar clustering is seen within CD4 T-helper cells, with Th17 cells clustering apart from other T-helper sub-types. We also see sub-type clustering within CD14 monocytes, with three distinct clusters: non-classical; intermediate and classical (see zoomed in section Fig 5). Therefore, as seen for the DNA methylation analysis in public data the marker loci appear to be able to provide a greater level of distinction than they were initially selected for. This speaks to the role of epigenetics in 'hard-wiring' cell lineage and regulating gene expression, and highlights the exciting possibility that DNA methylation could be explored to uncover previously unrecognised/identified immune cell sub-types.

Here we have further interrogated 11/1173 CpG sites identified in our initial discovery analysis of cell sorted immune cell populations from six healthy adult males—validating our observations in an independent cohort, and publicly available datasets (including both males and females). The public data also included a cohort of neonates demonstrating that the candidate loci held up in newborn samples too. We have not investigated whether differences are observed in individuals of varying ethnicity, although this would be an interesting avenue for further investigation. We have only looked at 11 loci; we believe that further investigation of the remaining sites with respect to their biological significance will likely reveal additional insights.

## Conclusion

In summary, this study highlights the value of mining publicly available data, the utility of DNA methylation as a discriminatory marker, the potential value of DNA methylation to provide additional insights into immune cell biology and the tantalising possibility that DNA methylation can be harnessed to reveal new aspects of cell biology including the identification of currently unrecognised/undistinguishable immune cell sub-types.

## Supporting information

**S1 Table. Annotation and other key information for all 1173 methylation CpG sites which were identified as being suitable cell-type markers.**
(CSV)

**S2 Table. Differential methylation statistics for pair-wise comparisons between cell populations of 11 CpG markers.**
(CSV)

**S3 Table. Differential expression statistics for pair-wise comparisons between cell populations of 11 gene transcripts.**
(CSV)

**S1 File.**
(TXT)

## Acknowledgments

We extend our appreciation to all the participants who volunteered for this study. Also, to Shaun Collings for volunteering to collect blood samples. Kylie Price and Sally Chappell (Malaghan Institute) for expert assistance with FACS. EpigenDX (USA) for running the pyrosequencing.

## Author Contributions

**Conceptualization:** Donia Macartney-Coxson, Miles C. Benton.

**Data curation:** Miles C. Benton.

**Formal analysis:** Jane Clapham, Miles C. Benton.

**Investigation:** Donia Macartney-Coxson, Miles C. Benton.

**Methodology:** Donia Macartney-Coxson, Alanna M. Cameron, Jane Clapham, Miles C. Benton.

**Supervision:** Donia Macartney-Coxson.

**Visualization:** Miles C. Benton.

**Writing – original draft:** Donia Macartney-Coxson, Alanna M. Cameron, Miles C. Benton.

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
