## [Decision Letter · Decision Letter 0]

6 Jul 2020

PONE-D-20-13841

DNA methylation in blood - potential to provide new insights into cell biology

PLOS ONE

Dear Dr. Benton,

Thank you for submitting your manuscript to PLOS ONE. After careful consideration, we feel that it has merit but does not fully meet PLOS ONE’s publication criteria as it currently stands. Therefore, we invite you to submit a revised version of the manuscript that addresses the points raised during the review process.

We look forward to receiving your revised manuscript.

Kind regards,

Osman El-Maarri, Ph.D

Academic Editor

PLOS ONE

Journal Requirements:

2. Please include your tables as part of your main manuscript and remove the individual files. Please note that supplementary tables (should remain/ be uploaded) as separate "supporting information" files

Reviewers' comments:

Reviewer's Responses to Questions

**Comments to the Author**

1. Is the manuscript technically sound, and do the data support the conclusions?

Reviewer #1: Partly

Reviewer #2: Yes

2. Has the statistical analysis been performed appropriately and rigorously? 

Reviewer #1: Yes

Reviewer #2: Yes

3. Have the authors made all data underlying the findings in their manuscript fully available?

Reviewer #1: Yes

Reviewer #2: Yes

4. Is the manuscript presented in an intelligible fashion and written in standard English?

Reviewer #1: No

Reviewer #2: Yes

5. Review Comments to the Author

Reviewer #1: Authors have gathered lot of data and have analyzed them rigorously but there are some aspects completly missing from the manuscript.

1. While gathering data for methylation, authors did not look for the expression data for the same cell types in publicly available databases

2. Author discussed about the statistical methods to analyze methylation data but no information was given for the batch effect correction, normalization and other methods for data processing. Also no information was found about the data downloaded from MARMAL-AID for example number of samples, age, population, gender. From t-SNE plot it seems many samples but no information was provided about these samples

3. The writing logical organization of this paper is poor and difficult to understand

Reviewer #2: In the manuscript entitled “DNA methylation in blood-potential to provide new insights into cell biology”, Macartney-Coxson et al., describe the results of a study aimed at identifying CpG-specific markers for different white blood cell types by mining data from a single publicly available data set consisting of Illumina HumanMethylation450-derived cell-specific DNA methylation signatures profiled across different leukocyte subtypes. The top two CpGs identified for five leukocyte subtypes were carried forward and validated via pyrosequencing in an independent data set. Moreover, gene expression was profiled in the subset of genes containing the aforementioned CpGs the same set of samples that underwent pyrosequencing, and used for comparisons of the expression profile across the five cell types. While the identification of lineage-specific DNA methylation markers is an area of great interest to the epigenetics research community given the now well-recognized potential for confounding by cell heterogeneity in EWAS of whole-blood or PBMCs and the fact that reference-based methodologies for deconvolution rely on such markers to enable accurate/reliable deconvolution estimates, the contributions of the research reported in this manuscript are incremental in comparison to what has already been published in this area. Despite this, the manuscript is well written and for the most part, easy to follow. Specific comments and suggestions for improvement are detailed below:

Major comments:

1. Above all, it is not clear what the unique contributions of this study are above and beyond work that has already been published in this area dating back to the 2012 publication of the Houseman method (PMID: 22568884) for mixture deconvolution; a method which relies on cell specific markers across leukocyte cell types, and the original Reinius paper (the discovery data set in this examination). Moreover, there are several key references (PMID: 29843789; PMID: 26956433; PMID: 27529193, to name a few) concerning the identification of white-blood-cell lineage markers that are missing and should be cited in the manuscript.

2. At the very least, the authors should consider using the GEO data set (GSE110555) as an additional validation data set. This GEO series consists of Illumina HumanMethylationEPIC data on isolated leukocyte subtypes profiled in different, healthy, non-diseased adults. While the array technology differs from the Reinius data set (450K), 90% of the CpGs on the 450K array are also contained on the EPIC array, so the vast majority of the ~1,000 cell specific markers identified in the discovery data set should be amenable to validation in an independent data set. The above mentioned data set is but one of several different publicly available data sets with leukocyte-specific methylation data profiled using either the 450K or EPIC data sets. Along these lines, it would be interesting to see if the markers identified here (identified in adult samples) hold up when examined in cord-blood derived leukocytes. GSE82084 was identified from a cursory search of GEO and contains 450K data on T cells, granulocytes, and monocytes isolated from cord-blood.

3. There is insufficient detail provided regarding the glmnet model used to identify cell-specific markers. What was the assumed response (and assumed distribution) in the fit of such models, what was the assumed random-effect, were the CpGs filtered in advance of fitting the glmnet model (e.g., removal/exclusion of sex-linked loci, cross-reactive probes, SNP associated CpGs, etc.), how were the tuning parameters for the elastic net model selected?

4. While a link is provided with the names/identities/annotations for the cell-specific markers identified in the discovery data set, in the opinion of this reviewer, such data should really be included as a supplementary table.

5. No limitations of the study are provided in the discussion. I would especially like for the authors to address the issue of the exclusive use of adult samples in their analysis and that such markers might not hold up in newborns or children. What is the race/ethnicity of the samples used in the discovery and validation data set and might that influence the results?

6. Were any formal statistical tests performed for the 11 CpGs in the validation samples? The results (Heatmap and Boxplot) are purely descriptive. From the looks of it, it seems that the results would be statistically significant, however a formal statistical method should be applied in this circumstance. On a related note, Figure S1, in my opinion, is more informative than the Heatmap used in Figure 2, as it conveys both central tendency and variability in the methylation/expression levels across the validation samples. The latter is not reflected in the Heatmap. As such, the authors should consider including Figure S1 as a Figure in the main manuscript.

7. It is mentioned numerous times in the manuscript that the analysis is “unsupervised”? How so? The analysis to identify cell-specific markers must be supervised in the sense that the identity of the cell type is used in the model as an independent variable.

8. It is mentioned in the conclusion that the results of this study can be “harnessed to reveal new aspects of cell biology included the identification of unrecognized/undistinguishable cell types”. How so?

6. PLOS authors have the option to publish the peer review history of their article (what does this mean?). If published, this will include your full peer review and any attached files.

Reviewer #1: No

Reviewer #2: No

---

## [Author Response · Author response to Decision Letter 0]

7 Sep 2020

Please note: for clarity reviewer comments are in italics, our responses are in normal text and additional text added to the manuscript as part of the revision is in red. 

Reviewer #1: Authors have gathered lot of data and have analyzed them rigorously but there are some aspects completely missing from the manuscript.

1. While gathering data for methylation, authors did not look for the expression data for the same cell types in publicly available databases

We selected 11 candidate loci for validation in an independent cohort of samples. DNA and RNA were extracted from the same sample simultaneously using the Qiagen AllPrep kit. DNA methylation was assayed using pyrosequencing and gene expression using real-time PCR. The results of the gene expression analyses are presented in Figure 2 in the original manuscript. In light of the reviewers comment we have also explored expression of these 11 genes in a publicly available RNASeq data ( GSE107011: https://www.ncbi.nlm.nih.gov/geo/query/acc.cgi?acc=GSE107011) dataset. A new section has been added to the manuscript to reflect this and a new Figure (Figure 5) presents a t-SNE analysis of the gene expression data which demonstrates clear separate of cell sorted immune cells with these 11 genes. 

Validation in publicly available data

RNA expression

To further explore expression of the 11 genes we selected for our validation of DNA methylation and RNA expression on independent samples we interrogated a publicly available RNAseq dataset, GSE107011 (Xu 2019). We extracted data for the same five cell populations (CD4+, CD8+, CD19+, CD56+ and CD14+) and then extracted expression data (TPM, transcripts per million) for each of the 11 genes. Figure 5 presents the data and illustrates the ability of mRNA expression from these 11 genes to clearly differentiate cell types. It is interesting to note that effector memory/terminal effector CD8+ cells (purple circles) are distinguished from central memory/naive CD8+ cells (purple squares) as well as a separation of Th17 CD4+ cells (yellow diamonds) from other CD4+ cells (yellow circles). Interactive versions of all figures are available online (https://sirselim.github.io/tSNE_plotting/).

2. Author discussed about the statistical methods to analyze methylation data but no information was given for the batch effect correction, normalization and other methods for data processing. Also no information was found about the data downloaded from MARMAL-AID for example number of samples, age, population, gender. From t-SNE plot it seems many samples but no information was provided about these samples

We thank the reviewer for providing the opportunity to provide more clarity. We have updated the methods section to better reflect the processing from ‘raw’ idat files through to analysis of methylation beta values. MARMAL-AID was the R package that was used to download the Reinius data, all applicable sample information is available at the GEO page (GSE35069, https://www.ncbi.nlm.nih.gov/geo/query/acc.cgi?acc=GSE35069). 

The Reinus data was downloaded using the R package MARMAL-AID. All applicable sample information is available at the GEO page (GSE35069, https://www.ncbi.nlm.nih.gov/geo/query/acc.cgi?acc=GSE35069). 

Raw intensity data (Illumina 450K idats) were loaded into R (R Core Team 2017) using the Bioconductor minfi package (Aryee 2014). Background correction and control normalisation was implemented in minfi. Probes were classed as failed if the intensity for both the methylated and unmethylated probes was <1,000. Any probe which failed in at least one sample, was removed from the entire dataset. We also removed all previously identified cross-reactive probes (Chen 2013), and 33 457 probes which we previously identified as aligning to the human genome greater than once (Benton 2017). All analyses were performed on beta values, calculated as the intensity of the methylated channel divided by total intensity including an offset ((methylated + unmethylated) + 100).

With respect to the t-SNE plot for 1173 CpG sites identified in our discovery analysis (Figure 1C) the points on the plot represent CpG sites not samples. We have made sure that the legend for each Figure which presents a t-SNE plot clearly states what the points represent (sample or CpG site as appropriate). This section is highlighted in the track changes version for each relevant figure legend. 

3. The writing logical organization of this paper is poor and difficult to understand

We respectively disagree with the reviewer but are more than happy to take editorial direction from the PLOSone team. We also note the comment from reviewer #2 that “…the manuscript is well written and for the most part, easy to follow”

Reviewer #2: In the manuscript entitled “DNA methylation in blood-potential to provide new insights into cell biology”, Macartney-Coxson et al., describe the results of a study aimed at identifying CpG-specific markers for different white blood cell types by mining data from a single publicly available data set consisting of Illumina HumanMethylation450-derived cell-specific DNA methylation signatures profiled across different leukocyte subtypes. The top two CpGs identified for five leukocyte subtypes were carried forward and validated via pyrosequencing in an independent data set. Moreover, gene expression was profiled in the subset of genes containing the aforementioned CpGs the same set of samples that underwent pyrosequencing, and used for comparisons of the expression profile across the five cell types. While the identification of lineage-specific DNA methylation markers is an area of great interest to the epigenetics research community given the now well-recognized potential for confounding by cell heterogeneity in EWAS of whole-blood or PBMCs and the fact that reference-based methodologies for deconvolution rely on such markers to enable accurate/reliable deconvolution estimates, the contributions of the research reported in this manuscript are incremental in comparison to what has already been published in this area. Despite this, the manuscript is well written and for the most part, easy to follow. Specific comments and suggestions for improvement are detailed below:

We thank the reviewer for their very constructive comments and believe that the additional analyses, clarification and discussion have significantly improved the manuscript. 

Major comments:

1. Above all, it is not clear what the unique contributions of this study are above and beyond work that has already been published in this area dating back to the 2012 publication of the Houseman method (PMID: 22568884) for mixture deconvolution; a method which relies on cell specific markers across leukocyte cell types, and the original Reinius paper (the discovery data set in this examination). Moreover, there are several key references (PMID: 29843789; PMID: 26956433; PMID: 27529193, to name a few) concerning the identification of white-blood-cell lineage markers that are missing and should be cited in the manuscript.

We thank the reviewer for this comment as it highlighted our need to clarify that our manuscript is not about immune cell deconvolution per se (as PMID: 26956433, 29843789, 27529193 [and many others] as indicated by the reviewer). We have modified the manuscript to provide additional clarity and acknowledge the plethora of papers exploring cell composition deconvolution as below, and that PMID.

Additional text in introduction:

Epigenetic marks, including DNA methylation, are increasingly recognised as potential discriminators of cell type (Salas 2018). This attribute has been utilised by a number of researchers to develop methods which correct for and/or deconvolute the variability introduced by cell mixtures in DNA methylation studies, particularly in blood samples (Houseman 2012, Salas 2018a, Kim 2016, Houseman 2016, Decamps 2020); a notable example - the so-called Houseman algorithm (Houseman 2012) - has been incorporated in to standard bioinformatic pipelines, including the R minfi package (Aryee et al 2014), for DNA methylation arrays. This behaviour of DNA methylation as a marker also suggests the possibility of such 'marks' revealing new aspects of biology - for instance it may highlight previously unrecognised immune cell populations. 

2. At the very least, the authors should consider using the GEO data set (GSE110555) as an additional validation data set. This GEO series consists of Illumina HumanMethylationEPIC data on isolated leukocyte subtypes profiled in different, healthy, non-diseased adults. While the array technology differs from the Reinius data set (450K), 90% of the CpGs on the 450K array are also contained on the EPIC array, so the vast majority of the ~1,000 cell specific markers identified in the discovery data set should be amenable to validation in an independent data set. The above mentioned data set is but one of several different publicly available data sets with leukocyte-specific methylation data profiled using either the 450K or EPIC data sets. Along these lines, it would be interesting to see if the markers identified here (identified in adult samples) hold up when examined in cord-blood derived leukocytes. GSE82084 was identified from a cursory search of GEO and contains 450K data on T cells, granulocytes, and monocytes isolated from cord-blood.

We are very grateful to the reviewer for this really constructive comment. We have now analysed 3 publicly available DNA methylation data sets (two using the Illumina EPIC arrays [one suggested by the reviewer] and the data from cord-blood derived leukocytes 9also highlighted by the reviewer) which used the earlier Illumina 450K arrays. This analysis has provided an additional, strong validation of the panel of CpGs identified by our initial analyses. We have modified the manuscript accordingly. 

Validation in publicly available data

DNA methylation

In order to further investigate the panel of 1173 CpG sites identified in our initial analysis we interrogated their methylation in 3 publicly available data sets. One, GSE82084, using the Illumina 450K platform (as per the Reinus data used in our discovery analysis) and two (GSE103541, GSE110554) using the more recent Illumina EPIC platform. Of the 1173 CpG sites 1025 were present on both platforms. The two EPIC studies performed DNA methylation analysis of cell sorted immune cell populations from adults (Salas 2018), whereas the 450K study looked at DNA methylation in cord blood from term and preterm newborns (de 2017). Figure 4 presents a 2D tSNE plot of all 1025 CpG sites which clearly shows separation of immune cells populations. It is interesting to note that the T cells of neonates (orange/red triangles) sit between the CD4+ and CD8+ T cells consistent with an undifferentiated state. 

Figure 4. tSNE plot of sorted-cells from 211 samples based on 1025 CpG sites overlapping between the three publicly available datasets. Points represent individual samples. 

3. There is insufficient detail provided regarding the glmnet model used to identify cell-specific markers. What was the assumed response (and assumed distribution) in the fit of such models, what was the assumed random-effect, were the CpGs filtered in advance of fitting the glmnet model (e.g., removal/exclusion of sex-linked loci, cross-reactive probes, SNP associated CpGs, etc.), how were the tuning parameters for the elastic net model selected?

We have provided more detailed information in the methods section addressing these comments.

Raw intensity data (Illumina 450K idats) were loaded into R (R Core Team 2017) using the Bioconductor minfi package (Aryee 2014). Background correction and control normalisation was implemented in minfi. Probes were classed as failed if the intensity for both the methylated and unmethylated probes was <1,000. Any probe which failed in at least one sample, was removed from the entire dataset. We also removed all previously identified cross-reactive probes (Chen 2013), and 33 457 probes which we previously identified as aligning to the human genome greater than once (Benton 2017). All analyses were performed on beta values, calculated as the intensity of the methylated channel divided by total intensity including an offset ((methylated + unmethylated) + 100).

Briefly, glmnet fits a generalized linear model via penalized maximum likelihood. The regularization path is computed for the lasso or elastic-net penalty at a grid of values for the regularization parameter lambda λ. The elastic-net penalty is controlled by α, and bridges the gap between lasso (α=1, the default) and ridge (α=0). The ridge penalty shrinks the coefficients of correlated predictors towards each other while the lasso tends to pick one of them and discard the others. The elastic-net penalty mixes these two; if predictors are correlated in groups, an α=0 tends to select the groups in or out together. We selected an alpha at the lower end of the range (0.05) to shift the elastic-net model more towards the penalised-regression (ridge regression), allowing us to retain more related features (CpG sites which share variance). For the GLMnet modelling we used cross-validation to determine the optimal value of regularization parameter λ with both minimum mean squared error (MSE) and minimum MSE + 1SE of minimum MSE. The optimal λ values were then used for predictor variable selection. 

4. While a link is provided with the names/identities/annotations for the cell-specific markers identified in the discovery data set, in the opinion of this reviewer, such data should really be included as a supplementary table.

In line with the reviewers comment we have provided this information as Supplementary information, in addition to the link to the Github repository where this, and further, information is provided. 

Supplemental Table S1: Annotation and other key information for all 1173 methylation CpG sites which were identified as being suitable cell-type markers. 

5. No limitations of the study are provided in the discussion. I would especially like for the authors to address the issue of the exclusive use of adult samples in their analysis and that such markers might not hold up in newborns or children. What is the race/ethnicity of the samples used in the discovery and validation data set and might that influence the results?

We thank the reviewer for this insight and in light of our re-analysis including the public data (see point 2 above) we demonstrate that in this initial exploration the selected markers perform equally as well across both adult and neonatal samples. We feel that the comment about ethnicity is outside of the scope of this research and think that this would be an excellent example of future refinements to this approach. We have modified the manuscript accordingly.

Discussion:

Figure 4 illustrates how well these 1025 CpG sites performed in additional, independent data. Furthermore, our initial analyses focused on samples from adults and as such we could not comment on their performance in neonates. However, one of the three public datasets we explored was from a study investigating DNA methylation in cord blood from term and preterm newborns. This clearly shows separation of immune cell sub-types isolated from neonates with the CpG markers we identified. 

…...

Here we have further interrogated 11/1173 CpG sites identified in our initial discovery analysis of cell sorted immune cell populations from six healthy adult males - validating our observations in an independent cohort, and publicly available datasets (including both males and females). The public data also included a cohort of neonates demonstrating that the candidate loci held up in newborn samples too. We have not investigated whether differences are observed in individuals of varying ethnicity, although this would be an interesting avenue for further investigation. We have only looked at 11 loci; we believe that further investigation of the remaining sites with respect to their biological significance will likely reveal additional insights. 

6. Were any formal statistical tests performed for the 11 CpGs in the validation samples? The results (Heatmap and Boxplot) are purely descriptive. From the looks of it, it seems that the results would be statistically significant, however a formal statistical method should be applied in this circumstance. On a related note, Figure S1, in my opinion, is more informative than the Heatmap used in Figure 2, as it conveys both central tendency and variability in the methylation/expression levels across the validation samples. The latter is not reflected in the Heatmap. As such, the authors should consider including Figure S1 as a Figure in the main manuscript.

Once again we thank the reviewer for this comment and suggestion. We have now included the boxplots within the body of the manuscript (Figure 3). In addition, we have added text to describe statistical analyses of the DNA methylation and RNA expression data for the 11 candidate loci, and provide the results of these as Supplementary tables. 

Added to methods:

Differential methylation and expression analyses were performed in R using the default student t-test. P values were adjusted using the Benjamini-Hochberg method. 

Added to results:

Figure 3. Boxplots illustrating DNA methylation and gene expression levels for all 11 gene investigated. Methylation and gene expression data for a given gene are in adjacent boxplots.

The eleven candidate loci were assayed by pyrosequencing in the 12 samples from the validation cohort. We observed a strong agreement with the expected discriminatory patterns of DNA methylation for all loci examined (Figures 2 and 3). Supplementary Table S2 presents pair-wise student T-test statistics for the DNA methylation data.

Given the role that DNA methylation plays in regulation of gene expression we also explored the mRNA levels of the 11 candidate loci. We investigated gene expression by QRTPCR in the 12 independent, validation samples. A clear differentiation between immune cells types at the gene expression level was observed for PARK2, POU2F2, DCP2, CD248, CD8A, SLC15A4, CD4A0LG and CD19 but not for FAR1, WIPI2, KLRB1 (Figures 2 and 3 ). Supplementary Table S3 presents pair-wise student T-test statistics for the gene expression data. 

Added to Supplementary Information

Supplemental Table S2: Differential methylation statistics for pair-wise comparisons between cell populations of 11 CpG markers.

Supplemental Table S3: Differential expression statistics for pair-wise comparisons between cell populations of 11 gene transcripts.

7. It is mentioned numerous times in the manuscript that the analysis is “unsupervised”? How so? The analysis to identify cell-specific markers must be supervised in the sense that the identity of the cell type is used in the model as an independent variable.

We take the reviewers point and to save confusion have removed the term ‘unsupervised’ from the manuscript and replaced it with “a machine learning approach” 

8. It is mentioned in the conclusion that the results of this study can be “harnessed to reveal new aspects of cell biology included the identification of unrecognized/undistinguishable cell types”. How so?

We have modified the manuscript to highlight the potential power of DNA methylation and aspects of our study which we believe suggest this possibility. We note that our exploration of publicly available data to further explore our DNA methylation and gene expression results (as suggested by both reviewers - many thanks) has improved our ability to provide this. 

Introduction:

This attribute has been utilised by a number of researchers to develop methods which correct for and/or deconvolute the variability introduced by cell mixtures in DNA methylation studies, particularly in blood samples (Houseman 2012, Salas 2018a, Kim 2016, Houseman 2016, Decamps 2020); a notable example - the so-called Houseman algorithm (Houseman 2012) - has been incorporated in to standard bioinformatic pipelines, including the R minfi package (Aryee 2014), for DNA methylation arrays. This behaviour of DNA methylation as a marker also suggests the possibility of such 'marks' revealing new aspects of biology - for instance it may highlight previously unrecognised immune cell populations. 

Discussion:

The potential of this approach is highlighted by our analyses of publically available data and 1025/1173 'candidate CpG sites' which overlapped between 450K and EPIC Illumina bead platforms. Figure 4 illustrates how well these 1025 CpG sites performed in additional, independent data. Furthermore, our initial analyses focused on samples from adults and as such we could not comment on their performance in neonates. However, one of the three public datasets we explored was from a study investigating DNA methylation in cord blood from term and preterm newborns. This clearly shows separation of immune cell sub-types isolated from neonates with the CpG markers we identified. It also suggests that such 'biomarkers' can potentially identify additional aspects of cell identity; for instance the T cells of neonates (orange/red triangles) sit between the mature CD4+ and CD8+ T cells consistent with an undifferentiated state. Furthermore, we also observed independent clustering of nucleated red blood cells from the same preterm newborns cohort. We believe these observations support the tantalising possibility that DNA methylation can be harnessed to reveal new aspects of cell biology including the identification of currently unrecognised/undistinguishable immune cell sub-types. 

We also investigated the expression of the 11 genes in a publicly available RNAseq dataset from immune cell sorted populations and saw a clear separation of the cell types with these 11 transcripts (Figure 5). In addition, the t-SNE analysis hints at the power of these 11 transcripts to provide a more nuanced separation of cell types. For example we observed distinct separation of CD8 T-cells into two clusters of sub-populations (Terminal Effector/Effector Memory and Central Memory/Naive). Similar clustering is seen within CD4 T-helper cells, with Th17 cells clustering apart from other T-helper sub-types. We also see sub-type clustering within CD14 monocytes, with three distinct clusters: non-classical; intermediate and classical (see zoomed in section Figure 5). Therefore, as seen for the DNA methylation analysis in public data the marker loci appear to be able to provide a greater level of distinction than they were initially selected for. This speaks to the role of epigenetics in 'hard-wiring' cell lineage and regulating gene expression, and highlights the exciting possibility that DNA methylation could be explored to uncover previously unrecognised/identified immune cell sub-types.

---

## [Decision Letter · Decision Letter 1]

14 Oct 2020

DNA methylation in blood - potential to provide new insights into cell biology

PONE-D-20-13841R1

Dear Dr. Benton,

We’re pleased to inform you that your manuscript has been judged scientifically suitable for publication and will be formally accepted for publication once it meets all outstanding technical requirements.

Kind regards,

Osman El-Maarri, Ph.D

Academic Editor

PLOS ONE

Additional Editor Comments (optional):

Reviewers' comments:

Reviewer's Responses to Questions

**Comments to the Author**

1. If the authors have adequately addressed your comments raised in a previous round of review and you feel that this manuscript is now acceptable for publication, you may indicate that here to bypass the “Comments to the Author” section, enter your conflict of interest statement in the “Confidential to Editor” section, and submit your "Accept" recommendation.

Reviewer #1: All comments have been addressed

Reviewer #2: All comments have been addressed

2. Is the manuscript technically sound, and do the data support the conclusions?

Reviewer #1: Yes

Reviewer #2: Yes

3. Has the statistical analysis been performed appropriately and rigorously? 

Reviewer #1: Yes

Reviewer #2: Yes

4. Have the authors made all data underlying the findings in their manuscript fully available?

Reviewer #1: Yes

Reviewer #2: Yes

5. Is the manuscript presented in an intelligible fashion and written in standard English?

Reviewer #1: Yes

Reviewer #2: Yes

6. Review Comments to the Author

Reviewer #1: Authors have addressed all my previous comments and have added the necessary information. Overall manuscript has communicates the work of the authors. I recommend to accept this paper.

Reviewer #2: The authors have adequately addressed the reviewer's comments and have incorporated the suggestions from the reviewers into their revised manuscript.

7. PLOS authors have the option to publish the peer review history of their article (what does this mean?). If published, this will include your full peer review and any attached files.

Reviewer #1: No

Reviewer #2: No

---

## [Editor Report · Acceptance letter]

26 Oct 2020

PONE-D-20-13841R1 

DNA methylation in blood - potential to provide new insights into cell biology 

Dear Dr. Benton:

I'm pleased to inform you that your manuscript has been deemed suitable for publication in PLOS ONE. Congratulations! Your manuscript is now with our production department. 

Kind regards, 

on behalf of

Priv.-Doz. Dr. Osman El-Maarri 

Academic Editor

PLOS ONE